# General route to design polymer molecular weight distributions through flow chemistry

Dylan J. Walsh [1], Devin A. Schinski[1], Robert A. Schneider[1] & Damien Guironnet [1]✉

The properties of a polymer are known to be intrinsically related to its molecular weight distribution (MWD); however, previous methodologies of MWD control do not use a design and result in arbitrary shaped MWDs. Here we report a precise design to synthesis protocol for producing a targeted MWD design with a simple to use, and chemistry agnostic computer-controlled tubular flow reactor. To support the development of this protocol, we constructed general reactor design rules by combining fluid mechanical principles, polymerization kinetics, and experiments. The ring opening polymerization of lactide, the anionic polymerization of styrene, and the ring opening metathesis polymerization are used as model polymerizations to develop the reactor design rules and synthesize MWD profiles. The derivation of a mathematical model enables the quantitative prediction of the experimental results, and this model provides a tool to explore the limits of any MWD design protocol.

[1] Department of Chemical and Biomolecular Engineering, University of Illinois Urbana–Champaign, Urbana, IL 61801, USA. ✉email: guironne@illinois.edu

A polymer's molecular weight distribution (MWD) impacts material properties such as processability, mechanical strength, and morphological phase behavior[1–6]. This correlation is general across all polymers and has motivated the development of many synthetic and process techniques. Much of the work regarding MWDs, has primarily focused on the development of polymerization methods to access polymers with narrow MWDs (referred to as controlled polymerization)[7,8]. Controlled polymerizations have revolutionized the synthesis of advance materials, however, the control they offer does not directly provide tunability for broad MWDs which are advantageous for many applications[9]. In fact, broad distributions remain a staple in industry[2,10]. For example, polyethylene produced by the Philipps catalyst has a dispersity >10, as this provides the ideal balance of fast processability and high mechanical strength[2,11]. Moreover, the latest improvements to polyolefins have been the tailoring MWDs for specific applications[2]. In addition, emerging areas of applications for thermoplastics (e.g., 3d printing) demand high mechanical performance while maintaining ease of processing, which can be achieved through tuning of the MWD[12,13]. While coarse MWD tuning is common practice, independent control over MW, MWD breadth, and MWD shape remains challenging. Thus, achieving high precision tailored MWDs remains a topic of high interest for studying material properties, as well as tailoring materials for specific applications.

Early approaches to engineered MWDs relied on blending distinct batches of polymers with known MW[1,2,5,14–18]. This simple approach requires synthesizing a large number of batches to reduce the multimodality of the final MWD (Fig. 1a). More recently, this has been streamlined by performing the polymerization with multisite catalysis or with cascade reactors[2]. However, multimodal MWD has been shown not to exhibit the same properties as a smooth MWD, making them unsuitable for several applications, e.g., macrophase separation[1,16]. This limitation motivated the development of synthetic methodologies for accessing polymers with tunable and smooth MWDs. Initial work in this area has focused on reducing the precision of controlled polymerizations which results in the smooth broadening of the MWD[19–27]. This strategy provides only little control over MW and no control over MWD shape. More recent work has sought to expand beyond broadening the MWD by utilizing reactor engineering strategies.

The first approach enables the skewing MWD profiles through the metered addition of a discrete initiating species into a controlled polymerization[3,28–30]. While successful, it is difficult to a-priori design the MWD of the polymer due to the complex polymerization kinetics caused by the constant variation in concentration of the growing chain and monomer[31]. The second approach implements flow reactors to synthesize narrow MWD polymer that accumulates into a reception vessel to build up a MWD profile[32–35]. However, the complex fluid mechanics and transport involved with performing a controlled polymerization in flow has limited the chemistry, molecular weight, and precision achievable[33,36–39]. The lack of reactor design rules necessitated significant empirical optimization of the process. Ultimately, the key limitation of prior MWD methodologies is that they do not use a design and result in arbitrary shaped MWDs.

Herein, we report a chemistry agnostic protocol to synthesize any MWD profile directly from a targeted design (Fig. 1b). The protocol consists of implementing a computer-controlled flow reactor to produce narrow MWDs (Fig. 2). Polymers with narrow MWDs accumulate in a collection vessel to construct any targeted MWD. The key feature of this approach is the ability to a-priori calculate the reactor flow rates needed to turn a MWD design into an actual polymer sample, which we describe as a design to synthesis protocol. The development of this protocol is supported by in-depth fluid mechanics and polymerization kinetics studies. These fundamental studies led us to establish the guiding principles for the design and operation of tubular flow reactors. We apply these principles for the synthesis of polymers with tailored MWDs. In addition, we developed a mathematical model that quantitatively predicts the experimental results, and enables us to explore the limit of any MWD design protocol that implements controlled polymerizations.

First, we describe the key reactor engineering concepts that enable the design of flow reactors. Second, we report on the implementation of this reactor design to the synthesis of MW sweep and unique MWDs. Lastly, we discuss the theoretical limits of MWD control via the development of a mathematical model.

## Results

**Fluid mechanics and reactor design rules.** The vast majority of laboratory-scale flow chemistry is performed in the laminar flow regime (low flow rates, small volume)[37,38,40,41]. Under these

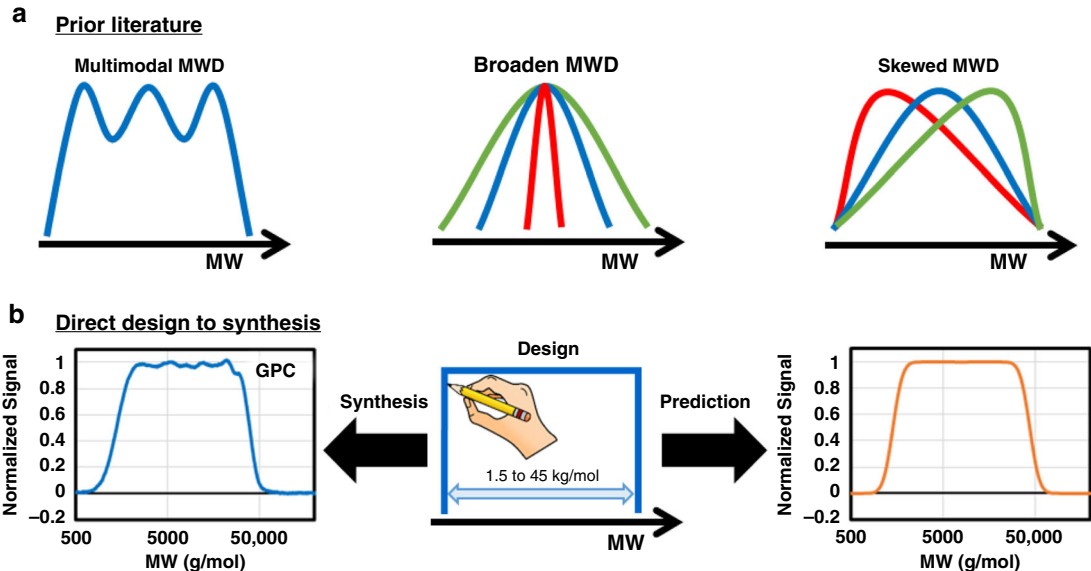

**Fig. 1 Overview of prior literature and this manuscripts methodology. a** Prior literature on MWD design. **b** Direct design to syntheisis methodology. MW molecular weight.

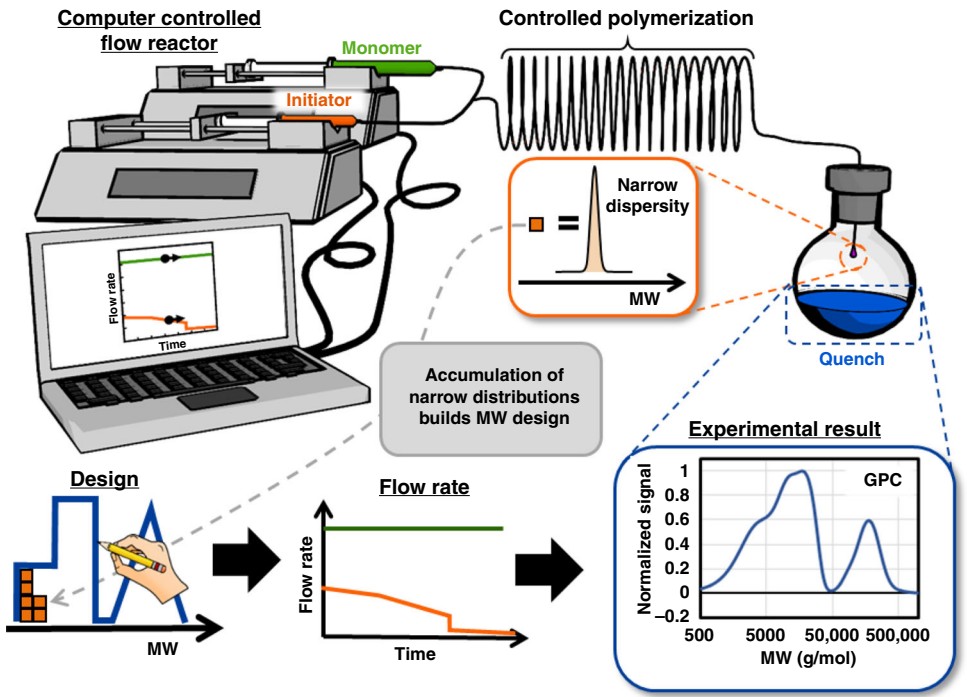

**Fig. 2 Overview of the flow reactor setup and methodology for MWD control.** The computer-controlled flow reactor generates narrowly dispersed polymer samples which accumulate in the collection vessel to build the target MWD. Reprinted (adapted) with permission from Macromolecules 2019, 52, 13, 4847–4857. Copyright (2019) American Chemical Society.

conditions, a parabolic flow velocity profile results in a wide distribution of residence times, and limited mixing (smooth streamlines) at the reactor inlet leads to inhomogeneity early in the flow process. For polymerizations, the distribution in residence times and/or poor mixing leads to the undesired broadening of MWDs[40,42–49]. Several strategies to circumvent these undesired effects have been developed over the years. The first one consists of performing the polymerization at very high flow rates[50,51]. This high flow rate consumes a significant amount of reagents making it expensive to operate and is only amenable to extremely fast polymerizations[52,53]. Another approach involves performing the polymerization in droplet flow, where plugs of non-reactive solvents or gases are placed in between plugs of polymerization mixtures[35,47,49,54]. The need to introduce pulses of an inert fluid into the reactor complicates the operation. Given the challenges of the previous methods, we elected to implement a simple tubular reactor that operates under laminar flow and achieves the essential plug flow via Taylor dispersion[55].

Taylor dispersion achieves a plug flow like behavior through diffusion and dispersion[56–59]. Specifically, the laminar flow will initially cause a solute pulse to stretch into a parabola, however, radial diffusion in conjunction with the radial velocity gradient will cause the homogenization of the concentration profile to yield a plug-like flow (Fig. 3c)[58,59]. This plug behavior is the enabling feature that makes it possible to achieve a narrow polymer MWD in laminar flow, as broad polymer MWD would be obtained otherwise due to initiators having different residence times. Taylor dispersion can also be leveraged more generally across flow chemistry where plugs are desired, such as high throughput synthesis[60,61].

To confirm Taylor dispersion's ability to generate plugs for polymerizations, we explored Talyor's original mathematical derivation in the context of our flow reactor[55,58,59]. The concentration profile that emerges from the derivation is functionally a normal distribution (Eq. 1) where the volume of the plug will depend on reactor radius (R), length (L), and flow rate (Q) (see Supplementary

Discussion for derivation). The derivation reveals that the plug volume will have a 2nd order dependency on reactor radius, 0.5 order dependency reactor length, and 0.5 order dependency on the flow rate (Eq. 2). In addition, the conditions under which Taylor dispersion is expected to apply for any tubular flow reactor set-up can be determined using in Supplementary Equation 41.

$$c(t) = \frac{M}{2\pi^{3/2}R^2\sqrt{D_{app}}\,t}e^{-\frac{(t-t_r)^2}{4D_{app}t_r/v_{z,avg}^2}}; \quad D_{app} = \frac{R^2v_{z,avg}^2}{48D_{ab}} \quad (1)$$

$$\textit{Plug volume} \propto Q\sigma \propto R^2\sqrt{LQ} \quad (2)$$

To validate this theoretical prediction, we performed several pulse tracer experiments by periodically introducing a UV absorbing initiator as a tracer to the ring-opening polymerization (ROP) of lactide (Fig. 3)[62]. We systematically varied the radius (0.0889–0.254 mm), length (7.6–15.2 m), and flowrate (63.4–267.5 μL/min) to determine their effect on the tracer pulse width. The tracer pulse width was determined by analyzing the polymerization mixture exiting the end of the tubular reactor using a GPC equipped with a RI and UV detector. The MW of the polymer produced during the tracer experiments was set to $M_n = 4500$ g/mol, and RI GPC data confirmed the stability of the process with a polymer $M_n = 4400 \pm 200$ g/mol and $M_w/M_n = 1.066 \pm 0.004$ throughout the entire experiment (see Supplementary Fig. 2). It is worth noting that the narrow MWD achieved in flow is very similar to batch data reported in the literature ($M_n = 4300$ g/mol, $M_w/M_n = 1.05$)[63]. Processing the UV tracer data gave rise to the predicted normal distribution of the concentration as a function of time. The standard deviation ($\sigma$) of these distributions, which is proportional to plug volume (Eq. 2), was calculated to have a second-order dependence on reactor radius, and half order dependency on length, matching the theoretical derivation (Fig. 4). However, a −0.86 order dependency on flow rate was measured (compared with the theoretical −0.5 dependency). Further tracer studies reviled, a −0.5 order dependency on

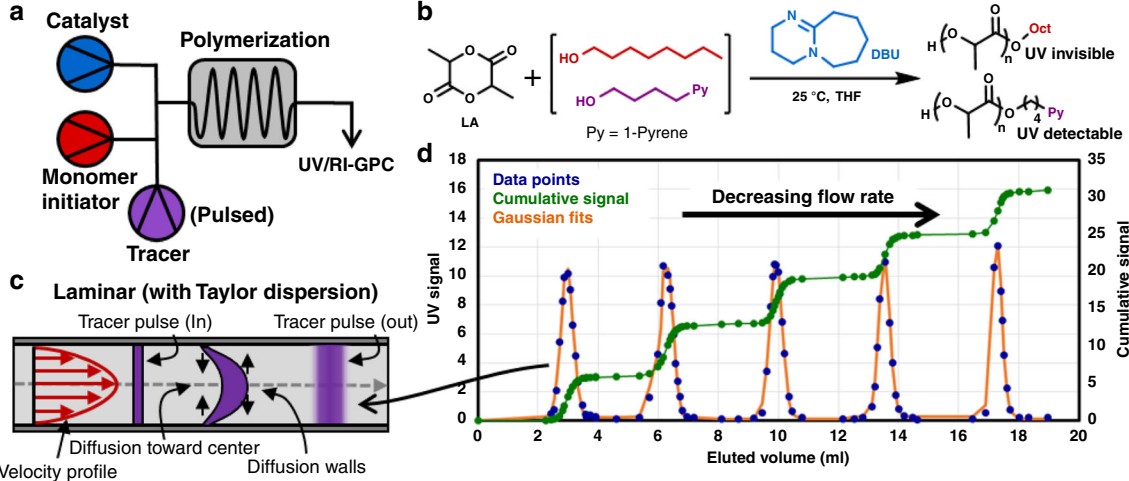

**Fig. 3 Reactor design, chemistry, fluid flow diagram, and data for polymer tracer experiments. a** Flow reactor configuration for the tracer experiments. **b** DBU catalyzed ROP of lactide with an octanol initiator. A UV detectable (266 nm) pyrene initiator will periodically be pulsed into the flow reactor as the tracer. **c** Depiction of laminar flow with Taylor dispersion. **d** Experimental data of tracer pulses for different flow rates in a single run.

flow rate when a small molecule tracer was used, and thus the deviation appears to be polymerization specific which will be the focus of a later study (see Supplementary Figs. 12–14). Validation of the theory enables Eq. (2) to be a quick and simple design rule for flow reactor design.

Controlled polymerizations, in general, require simultaneous initiation of all initiators in order to produce a polymer with narrow MWDs. However, under laminar flow, mixing of the monomer and initiator at the reactor's inlet is difficult due to the smooth streamlines[40,43–46]. To achieve mixing in this regime, custom static mixers are often implemented[37,64]. These mixers tend to be expensive and cause detrimental pressure drops for polymer synthesis. Thus, we elected to simply implement a Tee and rely on diffusion and dispersion to mix the solutions (Fig. 5a). For the initiation of the polymerization to occur homogeneously using diffusion and dispersion, the polymerization rate must be slow enough for mass transfer to homogenize the solutions. Therefore, experiments to determine the fastest maximum polymerization rate for our set-up was undertaken, while ensuring that we remain in a Taylor dispersion regime. We implemented the ring opening metathesis polymerization (ROMP) of an exo-norbornene type monomer in the flow reactor to probe the mixing efficiency[65–68]. ROMP was an ideal choice as it is extremely fast (90% conversion in 5s.), and the rate of polymerization can be tuned with the addition of pyridine over several orders of magnitude (see Supplementary Fig. 42).

The mixing efficiency was quantitatively probed by systematically increasing the 3-bromopyridne (Br-Py) to 3rd generation Grubbs catalyst (G3) ratio in a flow experiment, and analyzing the polymer produced with GPC (Fig. 5b, c). As expected, with very fast polymerization rates the heterogeneity of the reaction during the early stages of the polymerization caused a broadening of the MWD. However, with decreasing polymerization rates it was observed that the dispersity approaches the batch dispersity ($M_w/M_n = 1.03$) leveling off around 0.01 M/min[63]. Also, no additional effect on polymer dispersity from mixing was observed when the total flow rate or ratio of flow rates were changed (see Supplementary Figs. 19–25)[65,66]. This data provides a guide for designing flow reactors such that the rate of polymerization, or more generally, the rate of reaction does not exceed the mixing efficiency of the flow reactor implemented if precision is desired.

Overall, the above section provides the guiding principles to design flow reactors, which are broadly applicable to the field of

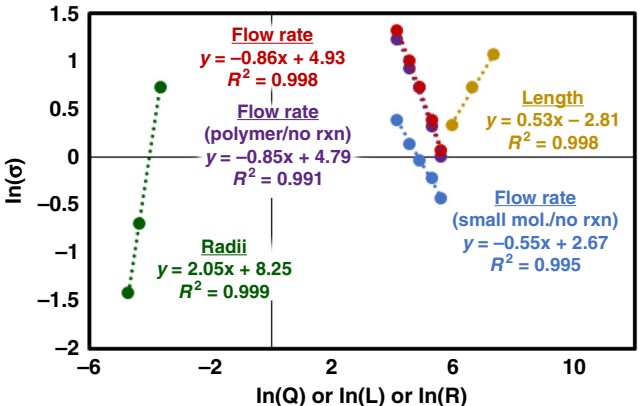

**Fig. 4 Processed tracer experimental results.** Green line is the best fit for $\sigma$ with different reactor radius ($R$). Tan line is the best fit for $\sigma$ with different reactor lengths ($L$). Red line is the best fit for $\sigma$ with different reactor flow rates ($Q$) while a polymerization is occuring. Purple line is the best fit for $\sigma$ with different reactor flow rates ($Q$) while no polymerization is occuring. Blue line is the best fit for $\sigma$ with different reactor flow rates ($Q$) using a small molecule tracer while no polymerization is occuring. ($\sigma$: standard deviation of the normal distribution of tracer exiting the tubular reactor).

flow chemistry. To leverage these reactor design principles for the engineering of MWD, we consider the MWD protocol consists of accumulating polymers with low dispersity. It then makes it advantageous to operate our reactor in a way that provides the smallest plug volume, since smaller plug volumes will result in a finer design resolution. We found that a reactor length of 762 cm and a reactor radius of 0.0889 mm or 0.127 mm achieves small plug while not generating too much backpressure for our syringe pumps. Additionally, the operation parameters (flow rates) should be on the order of 10's μL/min to achieve the appropriate residence time, given that the highest precision is achieved when the rate of polymerization is around 0.01 M/min. Using 10 mL syringes with this setup enables the synthesis of up to 500 mg of polymer with a single run. If larger quantities of sample are desired, multiple syringes pumps or continuous pumps can be used without altering the reactor design to produce multiple grams of material per day at high design resolution. If even larger amounts of polymer are needed, the radius of the reactor can be

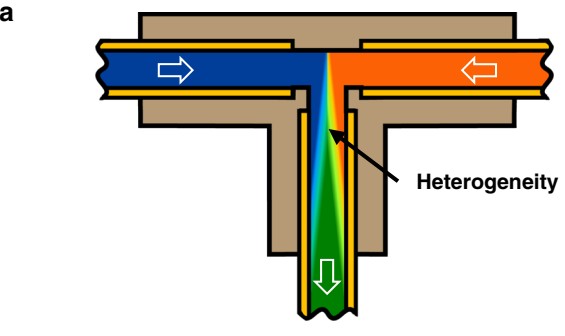

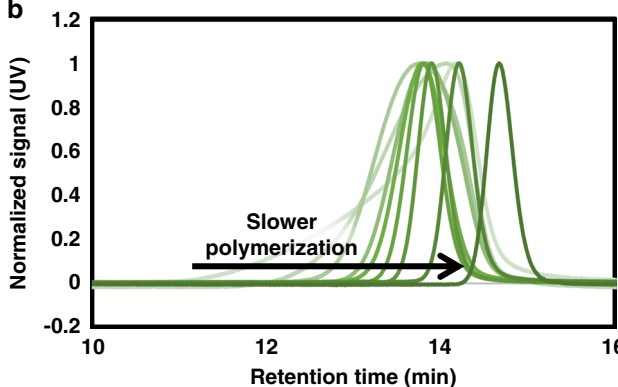

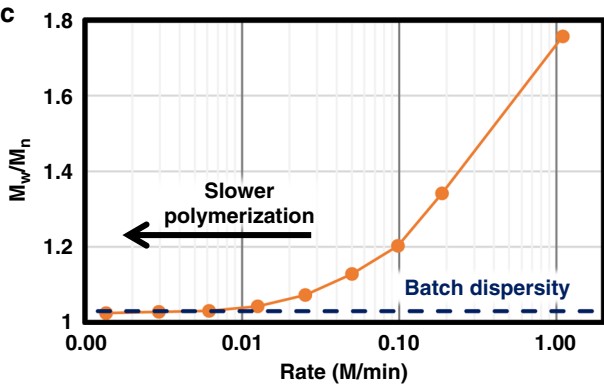

**Fig. 5 Diagram and experimental data for polymerization mixing experiments. a** Representation of the heterogeneity of the fluid at the entrance of the tubular reactor. **b** GPC traces for mixing experiment. **c** Dependancy of polymer dispersity on the rate of reaction.

increased, however, flow rate, reactor length, and polymerization rate may need to be adjusted to maintain the highest MWD design resolutions.

**MW sweeps in flow.** With the flow reactor design rules established, we sought to demonstrate the flow reactor's ability to produce precise quantities of polymer over a wide range of different MWs. We first targeted 6 different MWs (4, 16, 50, 300, and 1000 kg/mol) covering four orders of magnitude in a single flow experiment with ROMP. This MW sweep was achieved with a constant monomer flow rate, and by decreasing the G3 loading step-wise (Fig. 6). Br-Py (250 eq to G3) was added to the G3 solution to ensure that the rate of polymerization enabled sufficient time for mixing. The maximum flow rate was set so that the residence time allowed the reaction to reach complete conversion. It is interesting to note that by mixing the pyridine with the ruthenium, the polymerization becomes effectively zeroth order in initiator, and full monomer

consumption is achieved in the same amount of time for all MWs[65]. Upon performing and analyzing the MW sweep, 6 peaks were detected in the GPC chromatogram. The molecular weight of each peak matched the targeted MW confirming that the polymerizations were performed to completion, and the dispersities were low ($M_w/M_n = 1.02$–1.11). In addition, normal distributions were fitted to each of the 6 peaks and the area under each peak, which is proportional to the quantity of polymer formed, was measured to be $16.7 \pm 0.7\%$ (targeted value is 16.6%). Combining these two elements demonstrates the fine precision in molecular weight control achievable with our flow reactor.

We then demonstrated the generality of our reactor engineering strategy by performing the sec-BuLi initiated anionic polymerization of styrene[69]. The rate of polymerization was tuned with the addition of THF to the monomer mixture[70]. Repeating a MW sweep in a similar fashion as ROMP, establishes the ability of the flow reactor to achieve low MW dispersities ($M_w/M_n = 1.02$–1.11). Since we had use ROP to establish the design rule for the reactor we did not perform a MW sweep for the ROP of lactide. In contrast to ROMP and anionic polymerization, the ROP employs a catalyst and an initiator. This different composition led us to operate our reactor so that the degree of polymerization is altered by varying the conversion of monomer through the variation of the catalyst loading (kept below 70% to maintain low dispersity) while maintaining a constant residence time[55]. This approach of varying the degree of polymerization through variation of the monomer conversion is thought to be more compatible with other controlled polymerization (e.g. ATRP, anionic ROP, Coordinative Chain Transfer Polymerization (CCTP)).

**MWD design and modeling.** Having established the ability of the flow reactor to produce precise quantities of polymers with MWs spanning multiple orders of magnitudes, we sought to establish a design to synthesis protocol capable of converting any MWD profile into an actual sample. Building on the precision of the flow reactor, we developed a general mathematical procedure (see SI Figure 37) to calculate the process flow rates directly from the targeted shape; i.e. we can convert any design to synthesis. This straight forward design to synthesis workflow enables the a-priori design capability. To demonstrate this methodology we first targeted square MWDs (Fig. 7a) using the anionic polymerization of styrene. We produced four different MWDs with widths ranging from 1.5–5 kg/mol to 1.5–45 kg/mol. The MWD design is defined to be weight fraction on a log scale, as this is representative of a GPC chromatogram (see Supplementary Methods for more discussion). Figure 7a contains the four produced chromatograms, which qualitatively matches the targeted square MWDs (Table 1).

To further demonstrate the versatility of this methodology, we also synthesized two triangle MWDs using the anionic polymerization of styrene (see Supplementary Fig. 36). The two triangles span a MW range from 1.5 to 25 kg/mol with one design sloping right, and the other sloping left. Once again a qualitative match is observed with the targeted design and GPC data (see Supplementary Table 15 for tabulated data). Expanding beyond the anionic polymerization of styrene, we implemented ROMP for the synthesis of a target consisting of two discrete square distributions of different height and a triangle. This design is complex and covers a MW range from 2 to 400 kg/mol, challenging the methodology. As seen in Fig. 7b, the GPC trace shows the targeted MWD design features. An additional benefit of synthesizing non-normal distributions is the ability to easily generate large numbers of additional MWDs through blending (see Supplementary Fig. 37). Since the individual distributions are shaped and broad, there isn't a need to synthesize a large number of samples to achieve a smooth profile for the desired MWD design.

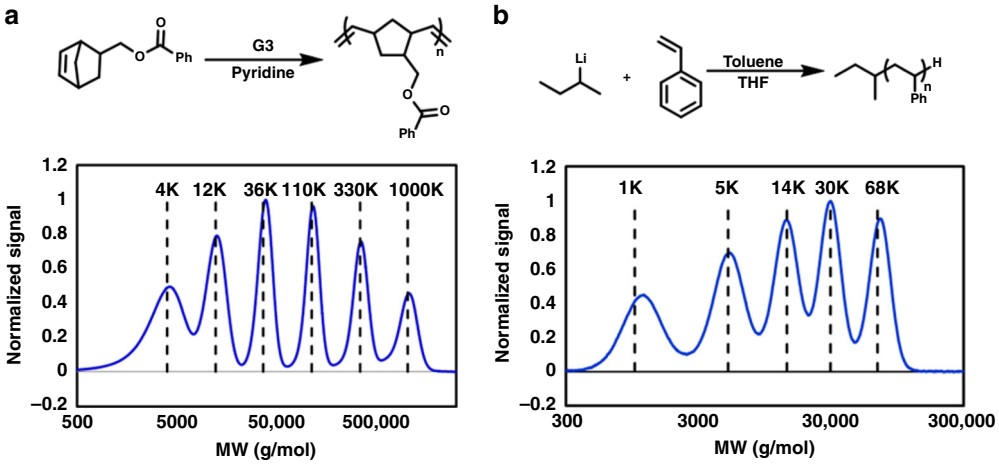

**Fig. 6 Chemistry and experimental data for MW sweep in flow reactor. a** ROMP of a norbornene type monomer with MW sweep chromatogram below it. **b** Anionic polymerization of styrene with MW sweep chromatogram below it. (dashed vertical lines are the target MWs).

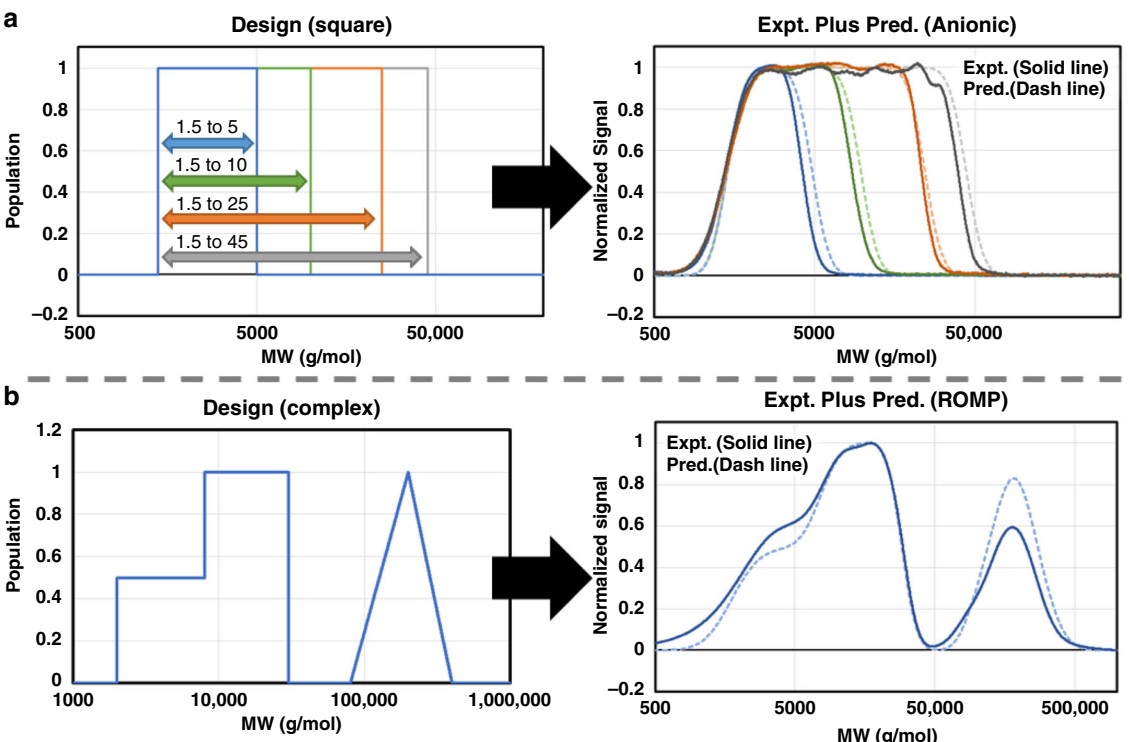

**Fig. 7 Designs, experimental data, and predictions for targeted MWD. a** (left) Design for the four targeted square MWDs. (right) GPC traces and predictions for the four square MWDs. **b** (left) Design for the targeted complex shape. (right) GPC trace and prediction for complex MWD.

**Table 1 Data for the synthesis of square MWD by anionic polymerization of styrene.**

| MW range (kg/mol) | $M_n$ (theory) (g/mol)[a] | $M_w/M_n$ (theory)[a] | $M_n$ (g/mol)[b] | $M_w/M_n$[b] |
|---|---|---|---|---|
| 1.5–5 | 2500 | 1.16 | 2370 | 1.16 |
| 1.5–10 | 3250 | 1.38 | 2920 | 1.37 |
| 1.5–25 | 4360 | 1.91 | 4030 | 1.93 |
| 1.5–45 | 5130 | 2.49 | 4630 | 2.50 |

[a]Values are from MATLAB code.
[b]Data obtained from GPC(THF) with respect to PS standards.

To further our analysis of the synthesized MWDs, a mathematical model was generated to predict the MWD. The mathematical model inputs the MWD design and generates a large number of polymer distributions, which are summed to produce the predicted MWD (Figs. 7 and 8). Polymer distributions are modeled by log-normal distributions, as is was observed that a normal distribution describes the MW sweeps GPC data well, and GPC data are plot on a log(MW) scale. Log-normal distributions are defined by two parameters: mean ($\mu_{LN}$) and standard deviation ($\sigma_{LN}$). The mean is defined by ln(MW), in which MW was determined from the design to ensure a linear spacing based on weight fraction. The standard deviation of the

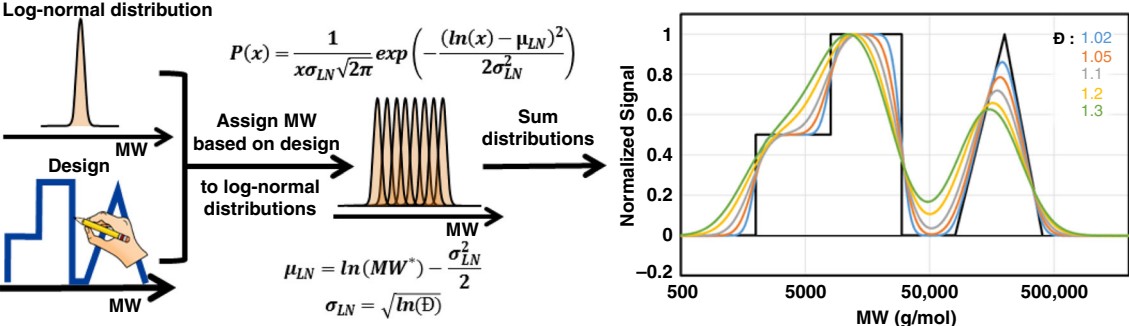

**Fig. 8 Overview of mathematical model for the prediction of MWD.** The mathematical model inputs the MWD design and generates a large number of log-normal polymer distributions with different MW which are summed to produce the predicted MWD. The is used to study the effect of dispersity on the final MWD. ($P(x)$: log-normal probability density function, $x$: MW scale; $\mu$: mean of log-normal distribution; $\sigma$: standard deviation of log-normal distribution; $MW^*$: linear spaced by weight fraction MW from targeted MWD design).

distribution was set by experimentally defined dispersities. As seen in Fig. 7 the model (dashed lines) produces a very close match to the experimental results (solid lines). Furthermore, the $M_n$ and dispersity of the sample can be calculated using the model and compared against the experimental values. The predicted $M_n$ and dispersity are in good agreement with experimental values (see Table 1 and Supplementary Table 15). Overall the quantitative match between the model and experiments establishes the precision of the synthesis and the validity of the model.

Finally, we use this mathematical model to investigate the effects of the dispersity of the polymer produced by the flow reactor on the final MWD. It becomes apparent that the resolution of the MWD is sensitive to the dispersity of the polymer produced. This is noted by the deviation of the predicted MWDs (colored lines) in comparison with the targeted MWD (black line) in Fig. 8. This result should be interpreted as the theoretical limit of the precision for any MWD design technique that relies on controlled polymerizations. In the context of flow reactors, this emphasizes the need for both chemistries that can produce very narrow MWD and a carefully designed reactor to ensure fluid mechanics does not erode the control of the polymerization.

## Discussion

We have developed a simple and robust procedure for converting any MWD profile directly into an actual sample by using a computer-controlled tubular flow reactor. This methodology consists of performing any controlled polymerization to produce low dispersity polymers in flow which accumulate into a collection vessel to generate the targeted MWD. Using this protocol, we prepared square, triangle, and a complex MWD profile, and quantitatively matched the experimental results to mathematical predictions. Fluid mechanics and polymerization kinetics provided the guiding principles for the construction and operation of the tubular flow reactor. The development of these design principles will significantly aid in the design and operation of flow reactors, which is broadly beneficial to the field of flow chemistry. We also anticipate that this simple, versatile, and precise methodology to directly synthesize any MWD will aid in fundamental material property studies, as well as, aid in the tuning of materials for specific applications.

## Methods

**General information**. All reactions were performed in oven-dried glassware under an argon atmosphere in an argon-filled glovebox ($O_2 < 2$ ppm, $H_2O < 0.5$ ppm) at room temperature unless otherwise specified. All solvents were dried using a solvent purification system. All commercially obtained reagents were used as received: 3-Bromopyridine {Aldrich, 99%}, Ethyl Vinyl Ether {Aldrich, 99%, store at −20 °C}, 1,8-Diazabicyclo[5.4.0]undec-7-ene (DBU) {Aldrich, 98%, store

at −20 °C}, Boric Acid {Aldrich, 99.5%}, D,L-lactide{Aldrich, 99%}, 1-Octanol {Aldrich, 99%}, SecBuLi {1.3 M sol. in cyclohexane/hexane (92/8) ACROS}. [(H$_2$IMes)(3-Br-py)$_2$(Cl)2Ru=CHPh], G3 was synthesized according to literature[67]. nor1 were synthesized according to literature[63]. Styrene {Aldrich} was pushed through an alumina column and distilled under vacuum prior to use.

**NMR experiments**. Nuclear Magnetic Resonance (NMR) spectra were recorded on a Bruker AVANCE III 500 MHz. Spectra are reported in ppm and referenced to the residual solvent signal: CDCl$_3$ ($^1$H 7.26 ppm, $^{13}$C 77.16 ppm), C$_6$D$_6$ ($^1$H 7.16 ppm, $^{13}$C 128 ppm), CD$_2$Cl$_2$ ($^1$H 5.32 ppm).

**Gel permeation chromatography**. Gel permeation chromatography (GPC) was performed using a Tosoh Ecosec HLC-8320GPC at 40 °C fitted with a reference column (6.0 mm ID × 15 cm), a guard column (6.0 mm ID × 4.0 cm × 5 μm), and two analytical columns (7.8 mm ID × 30 cm × 5 μm). The reference flow rate is 0.5 mL min$^{-1}$ while the analytical column is at 1.0 mL min$^{-1}$. THF (HPLC grade) was used as the eluent, and polystyrene standards (15 points ranging from 500 Mw to 8.42 million Mw) were used as the general calibration. An additional calibration was created for specifically for linear polylactic acid and only used for linear polylactic acid (10 points ranging from 500 Mw to 10,000 Mw).

**General procedure for MWD design**. In a typical experiment, the flow reactor is setup by first filling one syringe with a solution containing the monomer and a second syringe is filled with a initiator (an activator or inhibitor may be added to either syringe to adjust the rate of polymerizations). The syringes are then attached to the flow reactor and placed into the computer-controlled syringe pumps. The exit of the reactor was feed into a pot with a quenching reactant. A preprogram flow rate sequence then started. Upon completion, the final reaction mixture is analyzed by GPC and the final polymer is isolated by precipitation. A more detailed procedure can be found in the supporting information for each of the specific chemistry used in this manuscript.

## Data availability

All data generated or analyzed during this study are included in this published article (and its supplementary information files).

## Code availability

The MATLAB code used for predictions of MWD can be found in the supporting information.

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

## Acknowledgements

Major funding for the 500 MHz Bruker CryoProbe was provided by the Roy J. Carver Charitable Trust to the School of Chemical Sciences NMR Lab. The authors thank Umicore. for the generous gift of Grubbs Catalyst. The authors thank the NSF DMR 17-27605 for funding.

## Author contributions

D.W. contributed to the experimental design, preformed fluid mechanics experiments, preformed MWD experiments, produced the mathematical model to predict MWD, and wrote the paper. D.S. and R.S. assisted in fluid mechanic experiments. D.G contributed to the experimental design and wrote the paper.

## Competing interests

The authors declare no competing interests.
