## [Peer Review File · Nature Communications]

Reviewer #1 (Remarks to the Author):

Guironnet and co-workers report on the tailoring of molecular weight distributions for a variety of polymerizations through developing a mathematical model that enables predictive synthesis. This work appears to have been carefully performed and there are sufficient details in the manuscript and supporting information for reproducibility. This work is interesting and enables a more direct route for producing polymers with unique MWD profiles that will be useful for further studying macromolecular properties. This work can be accepted for publication after addressing these minor points.

- Dispersity is high or low while molecular weight distribution is broad or narrow. These descriptors have been used improperly throughout the manuscript
- The significant figures seem to not be consistent within datasets throughout the manuscript.

Reviewer #2 (Remarks to the Author):

The manuscript submitted herein by D. J. Walsh et al. describes the design & demonstration of a flow system that produces a set of arbitrary molecular weight distribution profiles by making & mixing a variety of polymers of narrow dispersity. The researchers take a bottom up approach that thoughtfully considers and investigates every aspect of the polymerization system with a focus on kinetics, reactor design, fluid dynamics, and how these critical aspects are interrelated. This paper will have broad appeal to people interested not only in custom MWD, but also polymerization kinetics in flow. The choice of experiments is well thought out and the authors take steps to isolate and understand fundamental variables. As an added aside, the supplementary info is very robust and will be helpful for those looking to reproduce this work. The manuscript is a nice example of the power of combining well thought out chemistry with a strong theoretical/mathematical background to create novel capabilities in polymer synthesis. For these reasons I recommend publication in Nature Communications after the minor revisions described below.

In table S4 & S5, & S6, the flow rates are mislabeled as mL/min instead of $\mu\text{L}/\text{min}$.

The work done in the reactor design & fluid dynamics section is very thorough and provides valuable fundamental knowledge to the community. I do believe, however, that the authors need to be more forthright about the disadvantages of their approach. For example, while Taylor dispersion holds for the conditions tested in Figure 4, both the tubing radii and flow rates tested fall in a narrow range of those actually applicable to the practicing flow chemist. It would help to move those actual values into the text so the reader doesn't have to dig through the SI to find them. It would also be helpful if the authors stated what deviations are expected when tubing radius increases above 0.254 mm or flow rate increases above the $\mu\text{L}/\text{min}$ range.

Also, as the authors state on page 11, the reliance on Taylor dispersion results in design rules that necessitate small tubing diameters ($<0.01''$) and slow flow rates (10's of μL s). This limits throughput and is a detriment if these reactions need to be scaled to a throughput necessary for applied studies. A sentence or two describing this limitation and the expected throughput of one of their systems would be welcome.

REVIEWER COMMENTS

Reviewer #1 (Remarks to the Author):

Guironnet and co-workers report on the tailoring of molecular weight distributions for a variety of polymerizations through developing a mathematical model that enables predictive synthesis. This work appears to have been carefully performed and there are sufficient details in the manuscript and supporting information for reproducibility. This work is interesting and enables a more direct route for producing polymers with unique MWD profiles that will be useful for further studying macromolecular properties. This work can be accepted for publication after addressing these minor points.

-Dispersity is high or low while molecular weight distribution is broad or narrow. These descriptors have been used improperly throughout the manuscript

- We have updated the writing according to the reviewer's suggestion.

-The significant figures seem to not be consistent within datasets throughout the manuscript.

- We have reviewed the manuscript in detail and ensured the appropriate significant figures are used.

Reviewer #2 (Remarks to the Author):

The manuscript submitted herein by D. J. Walsh et al. describes the design & demonstration of a flow system that produces a set of arbitrary molecular weight distribution profiles by making & mixing a variety of polymers of narrow dispersity. The researchers take a bottom up approach that thoughtfully considers and investigates every aspect of the polymerization system with a focus on kinetics, reactor design, fluid dynamics, and how these critical aspects are interrelated. This paper will have broad appeal to people interested not only in custom MWD, but also polymerization kinetics in flow. The choice of experiments is well thought out and the authors take steps to isolate and understand fundamental variables. As an added aside, the supplementary info is very robust and will be helpful for those looking to reproduce this work. The manuscript is a nice example of the power of combining well thought out chemistry with a strong theoretical/mathematical background to create novel capabilities in polymer synthesis. For these reasons I recommend publication in Nature Communications after the minor revisions described below.

In table S4 & S5, & S6, the flow rates are mislabeled as mL/min instead of $\mu\text{L}/\text{min}$.

- We appreciate the reviewer's thoroughness and have updated the labels.

The work done in the reactor design & fluid dynamics section is very thorough and provides valuable fundamental knowledge to the community. I do believe, however, that the authors need to be more forthright about the disadvantages of their approach. For example, while Taylor dispersion holds for the conditions tested in Figure 4, both the tubing radii and flow rates tested

fall in a narrow range of those actually applicable to the practicing flow chemist. It would help to move those actual values into the text so the reader doesn't have to dig through the SI to find them.

- We appreciate the reviewer's suggestion and have included the actual values in the main text.

It would also be helpful if the authors stated what deviations are expected when tubing radius increases above 0.254 mm or flow rate increases above the $\mu\text{L}/\text{min}$ range.

- We added a discussion on the bounds of the Taylor dispersion regime which should help to clarify the region of applicability and when the flow transitions into traditional laminar flow.

Also, as the authors state on page 11, the reliance on Taylor dispersion results in design rules that necessitate small tubing diameters ($<0.01''$) and slow flow rates (10's of μL s). This limits throughput and is a detriment if these reactions need to be scaled to a throughput necessary for applied studies. A sentence or two describing this limitation and the expected throughput of one of their systems would be welcome.

- We appreciate the reviewers comment about the throughput of our reactor design. The statements mentioned above (small tubing diameters ($<0.01''$) and slow flow rates (10's of μL s)) are under the pretense that small quantities (~ 500 mg) were desired (as in the case for making large libraries of materials). If there is a desire to produce larger quantities of material, the reactor design can be simply scaled up with the guidance of our reactor design principles. However, the key point of our manuscript was to develop a general high precision methodology for designing MWD, in which our methodology to convert a MWD design into a flowrate is not limited to a Taylor dispersion regime. We have now added a few sentences in the manuscript to clarify this point.
- To illustrate this point we decided to list how we would proceed if we needed to increase the throughput of our approach.
 - Multiple parallel reactors. The footprint of our reactor is small, and syringe pumps equipped with racked fitting multiple syringes are commercially available.
 - Switch from syringe pumps (which have a 10 ml syringe limit) to a continuous flow pump which is enable multiple grams of polymer to be produced in a day.
 - Use a larger reactor radius. This change requires additional considerations. First, the conditions under which the Taylor dispersion applies must be maintained by changing reactor length or flow rate or both. Second, you can expect the plug volume to increase with R^2 , so to maintain the same resolution the amount of polymer produced will also need to increase at least by squared amount as well.
 - Industrial level scale-up is also possible with this MWD design methodology, however, the flow reactor design would likely not be laminar flow regime but rather turbulent flow regime; which beyond the scope of this paper.